# Effect of Faint Road Traffic Noise Mixed in Birdsong on the Perceived Restorativeness and Listeners’ Physiological Response: An Exploratory Study

**DOI:** 10.3390/ijerph16244985

**Published:** 2019-12-07

**Authors:** Yasushi Suko, Kaoru Saito, Norimasa Takayama, Shin’ichi Warisawa, Tetsuya Sakuma

**Affiliations:** 1Department of Natural Environmental Studies, The University of Tokyo, Kashiwanoha 5-1-5, Kashiwa-shi, Chiba 277-8563, Japan; suko.yasushi@gmail.com; 2Office of Diversity Promotion, Forestry and Forest Products Research Institute, 1 Matsumoto, Tsukuba, Ibaraki 305-8687, Japan; hanri@ffpri.affrc.go.jp; 3Department of Human and Engineered Environmental Studies, The University of Tokyo, Kashiwanoha 5-1-5, Kashiwa-shi, Chiba 277-8563, Japan; warisawa@edu.k.u-tokyo.ac.jp; 4Department of Socio-Cultural Environmental Studies, The University of Tokyo, Kashiwanoha 5-1-5, Kashiwa-shi, Chiba 277-8563, Japan; sakuma@k.u-tokyo.ac.jp

**Keywords:** perceived restorativeness, skin conductance level, birdsong, natural sound

## Abstract

Many studies have reported that natural sounds (e.g., birdsong) are more restorative than urban noise. These studies have used physiological and psychological indicators, such as the skin conductance level (SCL) and the Perceived Restorativeness Scale (PRS), to evaluate the restorative effect of natural sounds. However, the effect of faint background noise mixed with birdsong on the restorativeness of birdsong has not been described yet. In the current experiment, we examined whether traffic noise affects the perceived restorativeness and the physiological restorativeness of birdsong in a low-stress condition using the SCL and the PRS. The scores of the PRS showed that birdsong significantly increased the perceived restorativeness of the place regardless of the car noise, but no significant difference was found between these two birdsongs. In contrast, physiologically, the birdsong without car noise decreased the participants’ SCL significantly more than the birdsong with car noise did. These results indicate that the SCL would be useful to detect the effect of background noise on natural sound when the noise is too low to affect the perceived restorativeness. This study highlights the importance of measuring the SCL besides assessing perceived restorativeness to describe the characteristics of restorative natural sound in future research.

## 1. Introduction

The restorative effects of natural sounds on human beings has been reported in many studies. Some of these natural sounds are birdsong, as well as water burbling in rivers, streams, and fountains [1,2,3]. In addition, research by Ma and Shu [4] revealed that these soothing sounds reduce fatigue and annoyance in people, and that auditory interventions may be more effective than visual stimulations in some circumstances under one particular condition. According to Li and Kang [5], birdsong in a forest at dawn and an ocean sound on a calm sunny day decreased the levels of the heart rate (HR), respiratory frequency (RF), and respiratory depth (RD) and increased the values of amplitude of the R-wave (ΔR), heart rate variability (HRV), electroencephalography alpha activity (α-EEG), and electroencephalography alpha activity (β-EEG). Among these natural sounds, birdsong was most frequently used as a representative of natural sound [2,3,4,6,7,8,9], and all the studies revealed that birdsong is more restorative than anthropogenic noise. 

These restorative natural sounds can be a useful resource for mental health care in modern society. Azuma and Inoue [10] implied that natural sounds would play an important role in improving the quality of life of the elderly in old-age homes as soundscapes are easier to manage than landscapes in such facilities. The benefits of natural sounds can also be useful in resting rooms for children with autism. According to Neo and Flaherty [11], such spaces, called “sensory rooms” in airports, are designed in such a way that children who play there are protected from loud noise. Adding natural sounds in such controlled environments would further improve the comfort provided by these spaces. De Coensel et al. [12] revealed that people would feel more pleasant when natural sounds were added to traffic noise.

However, the following limitations would have to be considered before implementing the above-mentioned studies into practice. Firstly, the types of natural sounds used in earlier studies are limited. Ma and Shu [4] concluded that birdsong and sound of water were the most preferred natural sounds based on the results of a laboratory experiment, but these sounds were not compared with other natural sounds, and instead were compared with artificial sounds like footsteps, ventilating noise, and traffic noise. Secondly, not every study contains adequate information on acoustic stimuli presented in its experiment. Hume and Ahtamad [6] used a realistic mixture of natural sounds and artificial sounds but did not provide detailed information on the acoustic characteristics of each sound-clip (18 types in total), although the mean value, the standard deviation, and the range of the sound pressure level across all the sound-clips were mentioned. Thirdly, the classifications of sounds used in some studies were exaggerated. In Medvedev et al. [8], each category (birdsong, ocean, construction, motorbike, airplane, and music) had only one type of sound, and the sounds of construction, motorbike, and airplane would understandably be unpleasant and people would not dare to listen to these sounds for stress recovery. For the same reason, Li and Kang’s experiment [5], which used four types of sounds, namely birdsong, ocean, street noise (an outdoor shopping street full of hurrying pedestrians and hawking), and traffic noise (an intersection at the peak hour in the afternoon on a sunny day), would not have been sufficient to investigate the characteristics of more restorative sounds among the sounds which have already been verified as restorative. However, it should be noted that the study surely confirmed that natural sounds were more restorative than anthropogenic noise. Additionally, these two studies [5,8] did not take silence as a control condition, which should have been necessary to examine whether listening to natural sounds was really more restorative than resting in silence.

On the basis of the above argument, an experiment where birdsong with the addition of small external disturbances, such as faint road traffic noise that people are normally exposed to in a park or in a forest to which they come to relax, would be more realistic and be worth investigating. Additionally, it would be necessary to assess the degree of restorativeness under silent conditions as the basis for comparison the effect of road traffic noise mixed with birdsong on the restorativeness and that of birdsong itself. This study aimed to investigate whether the presence or absence of road traffic noise affects the perceived restorativeness and the physiological restorativeness of birdsong. It is based on an exploratory methodology with a combination of physiological measurement and subjective evaluation. The research question of this study is as follows:
Does faint road traffic noise mixed in birdsong affect the restorativeness of the birdsong in terms of physiological evaluation even if the noise does not influence the perceived restorativeness of the place of experiment where people are exposed to the birdsong?

## 2. Materials and Methods 

### 2.1. Acoustic Stimuli and Normalization

Two types of two-minute long sound recordings recorded in a forest at dawn were used in this study. One is birdsong without road traffic noise, and the other is birdsong with road traffic noise that arises from the passage of three cars (hereon referred to as “car noise”). These sounds were recorded in Shiga highland, Japan. Before the main experiment, in which the sound recordings were produced using headphones to each subject, we conducted a preliminary experiment to adjust the volume to be moderate, where measuring the sound pressure level at ear canal entrance using a dummy head recording system. As a result, the equivalent continuous A-weighted sound pressure levels were 61 dB for “Birdsong without Car Noise”, and 63 dB for “Birdsong with Car Noise”, respectively. Figure 1 shows the octave band levels for the two reproduced sounds. It is seen that the birdsong has dominant components in 2 to 4 kHz bands, whereas the car noise raises the levels by around 10 dB in 250 to 500 Hz bands. 

From this preliminary experiment, we hypothesized that the car noise in the background reduces the restorativeness of birdsong physiologically even though it does not affect the perceived restorativeness of birdsong.

### 2.2. Physiological Measurement—Skin Conductance Level (SCL)

Skin conductance level (SCL) is the tonic level of electrical conductivity of skin [13], and is one of the metrics of electrodermal activity (EDA). EDA is the umbrella term used for defining autonomic changes in the electrical properties of the skin, and it is arguably the most useful index of changes in sympathetic arousal that are tractable to emotional and cognitive states. It is also the only autonomic psychophysiological variable that is not contaminated by parasympathetic activity [14]. EDA has been closely linked to autonomic emotional and cognitive processing, and EDA is widely used as a sensitive index of emotional processing and sympathetic activity [14]. In addition, compared to many other psychophysiological measures, EDA is relatively inexpensive to record [13]. There are mainly two types of EDA evaluation methods: skin conductance change (SCC) and skin potential activity (SPA). The SCL is classified in SCC, and according to Umezawa and Kurohara [15], the SCL is the most appropriate biofeedback index among the EDA indices.

Based on these facts, the SCL index was used in this study. It is measured by putting two electrodes on a participant’s distal phalange of the left forefinger and that of the left middle finger because the distal phalange site is recommended unless there are specific reasons for not using the distal site [13].

### 2.3. Psychological Measurement—Perceived Restorative Scale (PRS)

The Perceived Restorative Scale (PRS), which is based on the Attention Restoration Theory (hereon “ART”), advocated by Kaplan and Kaplan [16], is a questionnaire sheet to assess the restorative values of environments. The PRS was developed by Hartig and colleagues, and it consists of the following factors: “Being away” (being far from the ordinary present or routine aspects of one’s life, which a presumably necessary condition for restoration), “Fascination” (the degree to which an effortless attention is paid to particular contents and events driven by interest), “Coherence” (the ease with which one can organize and structure a scene), “Scope” (the scale of the domain in which the perceptual and organizational activity takes place), “Compatibility” (the match between the person’s goals and inclinations, environmental demands, and the information available in the environment for the support of intended and required activities), “Familiarity”, and “Preference” [17,18,19]. It comprises 26 items and participants made their responses on a 11-point scale to indicate the extent to which the given statement described their experience in the given natural sounds (0 = Not at all; 10 = Completely).

In this study, a Japanese version of the PRS, whose validity was confirmed by Shibata et al. [20], was used. This version was already used to investigate an attention restorative effect in the short-term staying of the on-site forest environment, and it was confirmed that such an effect in the forest environment was higher than the city environment [21]. Therefore, this Japanese version of the PRS would be applicable to this study.

### 2.4. Experiment Protocol

This experiment was conducted in a soundproof chamber in Kashiwa Campus, Department of Natural Environmental Studies, Graduate School of Frontier Sciences, at the University of Tokyo, Japan. There were 14 subjects including unpaid postgraduate students and members of staff. The mean age of the participants was 25.5 years (standard deviation (SD) = 3.11), consisting of six women and eight men. Several studies with physiological measurements were based on experiments with around 15 participants (e.g., 15 participants (four women and 11 men) in Horiuchi et al. [22]; 12 participants (12 men) in Lee et al. [23]). Therefore, the sample size of this exploratory study (14 participants) could also be considered as appropriate for the experimental design. In this study, the gender gap was not considered because we intended to know outcomes regardless of gender. We also limited the participants’ age span to around 25 years old because controlling the age span was considered to be useful to compare the results of this study with those of recent studies or future studies.

The participants were informed that the study was to investigate how people felt while listening to natural sounds, which was written on the cover page of the PRS questionnaire sheet. Therefore, the participants implicitly gave their informed consent by answering the questionnaire. During this experiment, participants wore a sleep mask and listened to natural sounds using headphones (ATH-PRO500MK2, Audio-Technica Corporation, Tokyo, Japan) so that they could completely concentrate on the sound (Figure 2). Figure 3 demonstrates the protocol of this experiment. Once the participants arrived at the place of the experiment, they were asked to answer the PRS questionnaire, in order to measure their psychological impressions towards the place of experiment as controls. In this study, the PRS was used to investigate whether the contents of acoustic stimuli changed the perceived restorativeness in the place of experiment. Next, a participant put two electrodes to measure the skin conductance level (SCL) on his/her distal phalange of the left forefinger and that of the left middle finger (Figure 4a), and wore a sleep mask and headphones. The SCL was measured by using Biosignalsplux, (PLUX Wireless Biosignals S.A., Lisbon, Portugal) (Figure 4b). After that, the sounds were presented in the following order: baseline (three min, silence), first sound (two min), interval (one min, silence), and second sound (two min). The length of the interval was determined based on the temporal difference between stimulus onset and recovery from a phasic increase in the SCL. According to the Handbook of Psychophysiology [13], the temporal difference between stimulus onset and the point of 50% recovery of a phasic increase in the SCL is around 30 s. Therefore, the interval in this experiment should be longer than 30 s to differentiate effects of the first sound from those of the other sound, but the interval should not be so long that it leaves the participants feeling bored. Thus, a one-minute interval could be considered as appropriate in this study. The order of presenting these sounds was changed for each participant to counterbalance the order effect. After finishing this step, each participant took the sleep mask off so that he/she could use his/her eyes to read the PRS questionnaire items and answer them. The participant listened to the natural sounds in the same order as he/she listened during the precedent SCL measurement period while filling in the PRS; “Birdsong without Car Noise” was being played while he/she was filling in the PRS to evaluate this sound, and “Birdsong with Car Noise” was being played while he/she was filling in the PRS to evaluate this sound. This was to compare the effect of the acoustic stimuli on the perceived restorativeness of the experimental place relatively. After that, the electrodes and the headphones were removed, and the participant was allowed to leave the room.

## 3. Results

### 3.1. The Perceived Restorativeness of the Experiment Environment

The score of the PRS of “Birdsong without Car Noise” and “Birdsong with Car Noise” were analyzed to examine to what extent the contents of the acoustic stimuli changed the perceived restorativeness of the experiment environment. The mean value and the standard error of the score of each attribute of the PRS in all the participants (*n* = 14) was evaluated. According to the Kruskal–Wallis test of each attribute of the PRS by the type of acoustic stimuli, significant differences between the three conditions (“Birdsong without Car Noise,” “Birdsong with Car Noise,” and “Control (Silence)”) were found in Scope only. The Steel–Dwass test was conducted in this attribute for the multiple comparisons and the test revealed that “Birdsong without Car Noise” and “Birdsong with Car Noise” made the score of Scope significantly higher than “Control (Silence)” did, although no significant difference was found between the two sounds (Table 1).

### 3.2. Results in Each PRS Attribute

#### 3.2.1. Being Away

The mean value of Being away was 8.36 with the acoustic stimuli of “Birdsong without Car Noise,” which was higher than 5.52 with “Birdsong with Car Noise,” and 6.32 with “Control (Silence).” According to the Kruskal–Wallis test, there was no significant difference between the three conditions (*p* = 0.0588, see Table 1), and the effect size (Cramer’s V) in this test was 0.45.

#### 3.2.2. Fascination

The mean value of Fascination was 8.06 with the acoustic stimuli of “Birdsong without Car Noise,” which was higher than 6.66 with “Birdsong with Car Noise,” and 6.96 with “Control (Silence).” No significant difference was found between “Birdsong without Car Noise,” “Birdsong with Car Noise,” and “Control (Silence).” According to the Kruskal–Wallis test, there was no significant difference between the three conditions (*p* = 0.2026), and the effect size (Cramer’s V) in this test was 0.34. 

#### 3.2.3. Coherence

The mean value of Coherence was 5.63 with the acoustic stimuli of “Birdsong without Car Noise,” which was higher than 5.00 with “Birdsong with Car Noise,” and 5.48 with “Control (Silence).” No significant difference was found between “Birdsong without Car Noise,” “Birdsong with Car Noise,” and “Control (Silence).” According to the Kruskal–Wallis test, there was no significant difference between the three conditions (*p* = 0.7844), and the effect size (Cramer’s V) in this test was 0.13.

#### 3.2.4. Scope

The mean value of Scope was 8.08 with the acoustic stimuli of “Birdsong without Car Noise,” which was higher than 5.35 with “Birdsong with Car Noise,” and 3.20 with “Control (Silence).” A significant difference was found between “Birdsong without Car Noise” and “Birdsong with Car Noise” (*p* < 0.01), between “Birdsong without Car Noise” and “Control (Silence)” (*p* < 0.01), and between “Birdsong with Car Noise” and “Control (Silence)” (*p* < 0.01). According to the Kruskal–Wallis test, there was a significant difference between the three conditions (*p* = 0.0001), and the effect size (Cramer’s V) in this test was 0.89.

#### 3.2.5. Compatibility

The mean value of Compatibility was 6.72 with the acoustic stimuli of “Birdsong without Car Noise,” which was higher than 4.66 with “Birdsong with Car Noise,” and 5.48 with “Control (Silence).” A significant difference was found between “Birdsong without Car Noise” and “Birdsong with Car Noise” (*p* < 0.05), whereas no significant difference was found between “Birdsong without Car Noise” and “Control (Silence),” or between “Birdsong with Car Noise” and “Control (Silence).” According to the Kruskal–Wallis test, there was a significant difference between the three conditions (*p* = 0.0333), and the effect size (Cramer’s V) in this test was 0.49.

#### 3.2.6. Familiarity

The mean value of Familiarity was 4.00 with the acoustic stimuli of “Birdsong without Car Noise,” which was higher than 3.90 with “Birdsong with Car Noise,” and 3.10 with “Control (Silence).” No significant difference was found between “Birdsong without Car Noise,” “Birdsong with Car Noise,” and “Control (Silence).” According to the Kruskal–Wallis test, there was no significant difference between the three conditions (*p* = 0.1169), and the effect size (Cramer’s V) in this test was 0.39.

#### 3.2.7. Preference

The mean value of Preference was 6.80 with the acoustic stimuli of “Birdsong without Car Noise,” which was higher than 4.40 with “Birdsong with Car Noise,” and 4.50 with “Control (Silence).” A significant difference was found between “Birdsong without Car Noise” and “Birdsong with Car Noise” (*p* < 0.01), between “Birdsong without Car Noise” and “Control (Silence)” (*p* < 0.05), whereas no significant difference was found between “Birdsong with Car Noise” and “Control (Silence).” According to the Kruskal–Wallis test, there was no significant difference between the three conditions (*p* = 0.0564), and the effect size (Cramer’s V) in this test was 0.45.

#### 3.2.8. Overall Trend

Among the seven attributes of the PRS, the scores of Scope under the condition of “Birdsong without Car Noise” and “Birdsong with Car Noise” were significantly higher than those under the condition of “Control,” but there was no significant difference between “Birdsong without Car Noise” and “Birdsong with Car Noise.” The other attributes did not show any significant difference between these three conditions. 

### 3.3. Physiological Response

The participants’ SCL value while listening to “Birdsong without Car Noise” and “Birdsong with Car Noise” was analyzed. This value corresponds to the extent to which these acoustic stimuli affect the participants’ physiological response, namely, the degree of the sympathetic nerve activity, which corresponds to the degree of their stress. Figure 5a shows the SCL changes of Subjects 1 to 7, who listened to “Birdsong without Car Noise” first followed by “Birdsong with Car Noise,” and Figure 5b shows that of Subjects 8 to 14, who listened to these sounds in the reversed order. The pattern of change in the SCL differed from person to person; for example, the SCL of Subject 1 fluctuated more than those of Subjects 7 and 13. As these graphs show, the SCL varies among individuals in terms of absolute values and changing patterns, therefore relative values of the SCL were calculated in the following way to compare the physiological restorativeness of the two sounds. First, the mean SCL value in each 10 s during the two-minute listening periods of every participant was calculated. Second, this value in the 10 s periods were normalized by the mean value during the first 10 s (0–10 s) in every participant. This normalized value is the relative SCL value in each participant. Next, the mean value and the standard error of this relative value of all the participants (*n* = 14) in each 10 s period was calculated.

According to Welch’s *t*-test, no significant differences were found between “Birdsong without Car Noise” and “Birdsong with Car Noise” during 10–20 s, 20–30 s, 30–40 s, and 110–120 s. By contrast, the relative SCL value under the former condition was significantly lower than what was under the latter condition from 40 to 110 s. These outcomes were based on *p*-values, which is the probability of obtaining the observed results of a test, assuming that there is no difference in the mean values of the relative values of the SCLs between the two conditions (see Table 2). “Birdsong without Car Noise” kept the relative value of the SCL lower than 1.0 throughout this period, whereas “Birdsong with Car Noise” increased the relative value and kept it higher than 1.0 (see Figure 6 and Table 2). The effect size (r) in each ten-second period was as follows: 0.028 (10–20 s), 0.111 (20–30 s), 0.201 (30–40 s), 0.465 (40–50 s), 0.517 (50–60 s), 0.527 (60–70 s), 0.504 (70–80 s), 0.495 (80–90 s), 0.510 (90–100 s), 0.450 (100–110 s), and 0.333 (110–120 s).

The standard errors of “Birdsong with Car Noise” increased 50 s after the acoustic stimulus onset, whereas “Birdsong without Car Noise” did not.

## 4. Discussion

The result of the PRS evaluation indicates that “Birdsong without Car Noise” did not differ from “Birdsong with Car Noise” in terms of the perceived restorativeness, whereas these two conditions had significantly higher scores for “Scope” than “Control.” According to Kaplan and Kaplan [16] and Shibata et al. [20], the attribute “Scope” is considered as being related to the degree of restorativeness of an environment. Therefore, the birdsongs may have enhanced the perceived restorativeness of the place of experiment (the soundproof chamber) regardless of the presence of the faint background car noise (Note: The attribute “Scope” was created as a subscale of “Extent” by Shibata et al. [20]). These results imply that the acoustic stimuli of birdsongs in a forest would be as restorative as an actual forest environment [21] or a picture of natural settings [20]; although, the presence or absence of slight background noise may not affect the perceived restorativeness of sound in nature. However, the effect of the acoustic stimuli on the perceived restorativeness of the experimental place might have been underestimated in this limited experimental design because the PRS contains some questions which are not readily associated with acoustic stimuli. Nevertheless, the PRS was developed to measure the perceived restorativeness of one particular environment, and therefore it is not necessarily specific to assessing landscapes. Although many eyesight-oriented studies have used the PRS, there have also been on-site studies in which the PRS was used to evaluate the perceived restorativeness of the environment which participants experienced with the five senses. Therefore, it would be acceptable to use the PRS in a hearing-oriented study. In this study, using the PRS helped examine the validity of its outcome by comparing the result with those of recent studies which used the PRS. Besides, this study aimed to investigate whether the perceived restorativeness of the environment with different acoustic stimuli differ on the presupposition that the stimuli of the other four senses (including the eyesight) are the same. Therefore, the result of this study—the acoustic stimuli of birdsong changed the perceived restorativeness of the environment, but that of faint road traffic noise did not—was compatible with our intention to use the PRS.

The result of the SCL measurement indicates that “Birdsong without Car Noise” inhibited the participants’ sympathetic nerve activity during the listening period, whereas “Birdsong with Car Noise” did not. This result could be interpreted as follows: The presence of the faint background car noise may have prevented birdsong from being restorative. Besides, the difference in standard errors of the relative values of the SCL between the two conditions may imply that this effect of car noise would appear at least 50 s after birdsong begins, but the degree of this effect would differ from person to person. Mainly, the individual difference of the permittable range of artificial noise would be more extensive than that of natural components. It is contradictory to the result of the PRS score, which showed that “Birdsong without Car Noise” and “Birdsong with Car Noise” did not differ in terms of the perceived restorativeness. This contradiction could be explained as follows: If background noise is too low to be perceived, it does not affect the perceived restorativeness of birdsong; however, it still reduces the restorativeness of the birdsong in terms of the SCL, namely, a physiological response. Therefore, such a noise would have an effect on restoration physiologically when people are listening to natural sounds, even if the noise is too slight to be perceived subjectively. This finding deepened the knowledge on the effect of background noise on the restorativeness of natural sounds because most of the preceding studies used exaggerated acoustic stimuli in experiments and did not examine effects of such a minor difference (e.g., [4,5,8]). The finding of this study also indicates that several studies which exposed participants to each sound for less than 50 s might have overlooked the effect of artificial components (e.g., 30-s sound excerpts were evaluated subjectively in Axelsson et al. [24]; eight-second sound excerpts were assessed subjectively and physiologically in [6]). In that sense, measuring the SCL for more than 50 s would be useful to detect the negative effect of background noise on the restorativeness of birdsong when the noise cannot be perceived subjectively. Therefore, it would be recommended to use physiological measurements like the SCL besides subjective measures to describe the restorativeness of sounds in nature.

This study has some limitations. Regarding the restorative process during the experiment, we cannot exclude the possibility that each one-minute silent interval also affects the degree of restoration. Besides, no stressful precedent condition was provided. The birdsong with or without car noise could have restored the participants in different ways if the participants had been under stress before the listening period. Wearing a sleep mask would also affect the degree of restorativeness by making a participant feels sleepy while listening to the sounds. Some of the questions in the PRS would also not have been easily affected by acoustic stimuli. To overcome the above shortcomings, further research on the subject should be conducted with more respondents across a broader age span. 

## 5. Conclusions

Although many studies revealed that natural sounds, especially birdsong, are more restorative than artificial sounds, the effect of faint background noise on the restorativeness of birdsong had not been described in detail yet. This study investigated whether such noise affects both the perceived restorativeness and the physiological restorativeness of two-minute birdsongs by measuring the PRS and the SCL. The experiment revealed that “Birdsong without Car Noise” made the SCL significantly lower than “Birdsong with Car Noise,” whereas these sounds were not significantly different in terms of the perceived restorativeness. These facts imply that the SCL would be useful to detect the unconscious effect of background noise on natural sounds when the effect cannot be found by subjective evaluation. This study highlights the importance of measuring the SCL besides the assessment of perceived restorativeness to describe the characteristics of restorative natural sounds in future research. The outcome of this exploratory study can also be applied in medical settings, where staff members, especially surgeons and surgical nurses, are subjected to stress every day. Measuring surgeons’ or nurses’ SCL during surgery would be useful to detect their stress levels due to auditory disturbances as their negative effects are not perceived subjectively. This detection can help in creating a more efficient environment in medical settings. 

## Figures and Tables

**Figure 1 ijerph-16-04985-f001:**
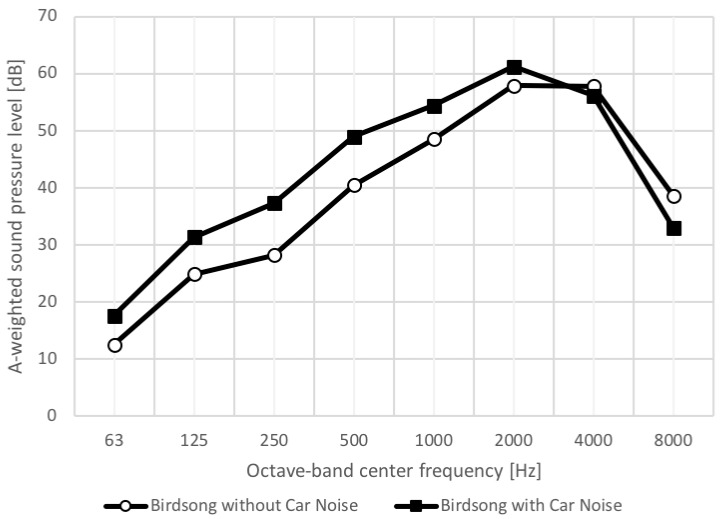
Frequency characteristics of equivalent continuous A-weighted sound pressure levels for the two acoustic stimuli.

**Figure 2 ijerph-16-04985-f002:**
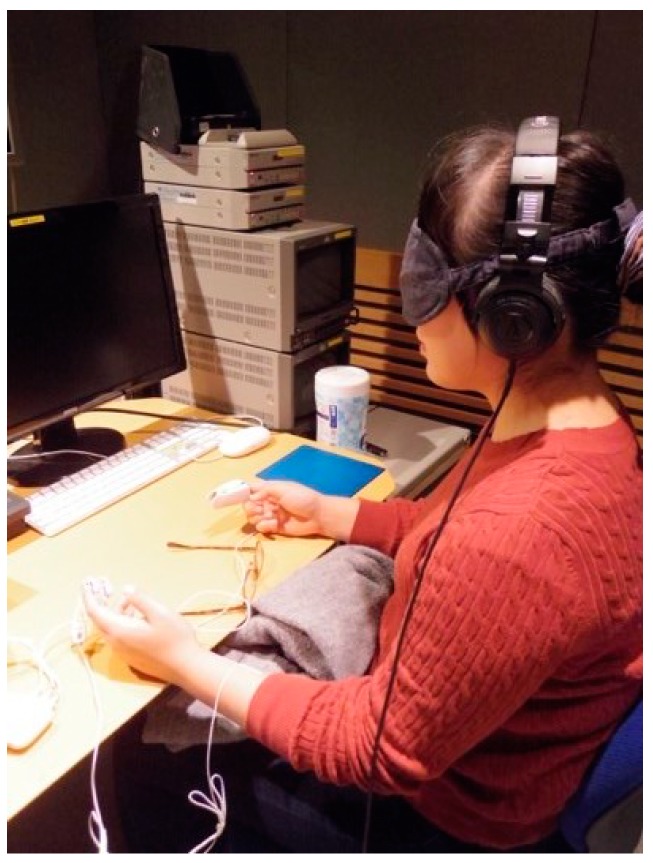
Participant in a soundproof room.

**Figure 3 ijerph-16-04985-f003:**
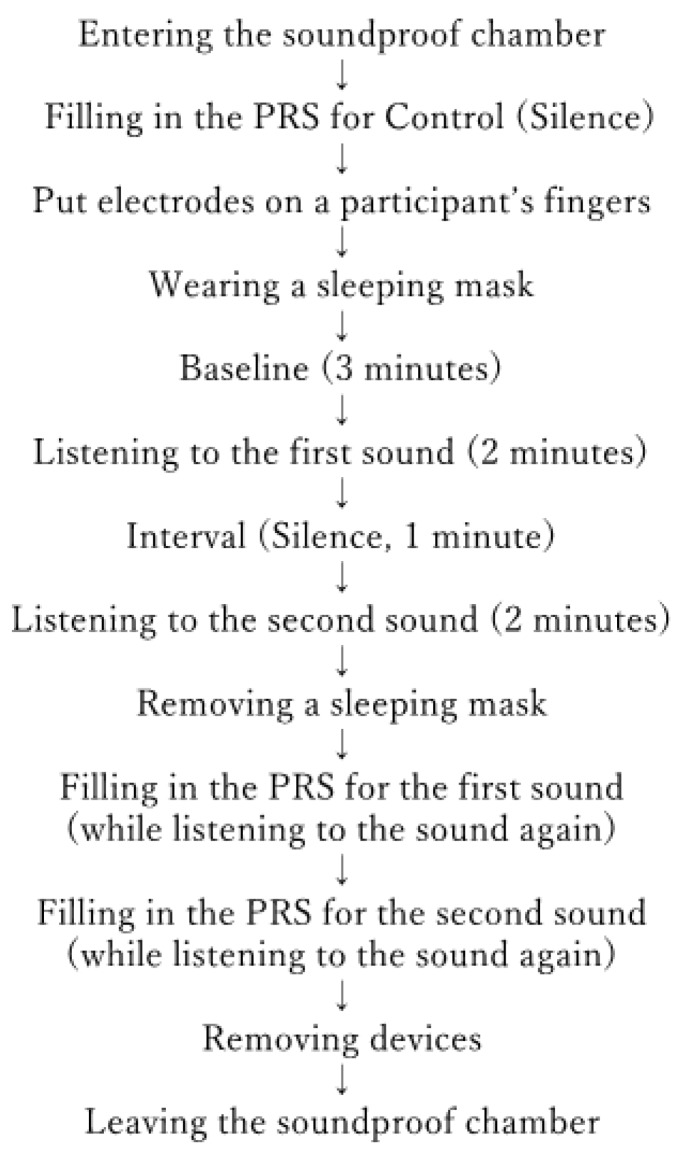
Experiment protocol.

**Figure 4 ijerph-16-04985-f004:**
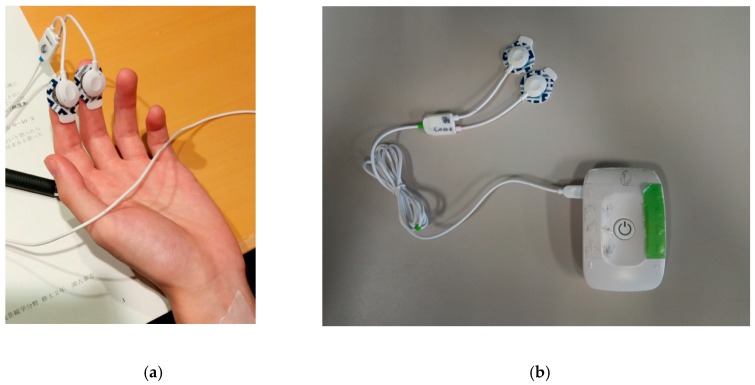
(**a**) The electrodes for the skin conductance level (SCL) measurement on the left hand. (**b**) The biosignal sensors (Biosignalsplux) used in this experiment.

**Figure 5 ijerph-16-04985-f005:**
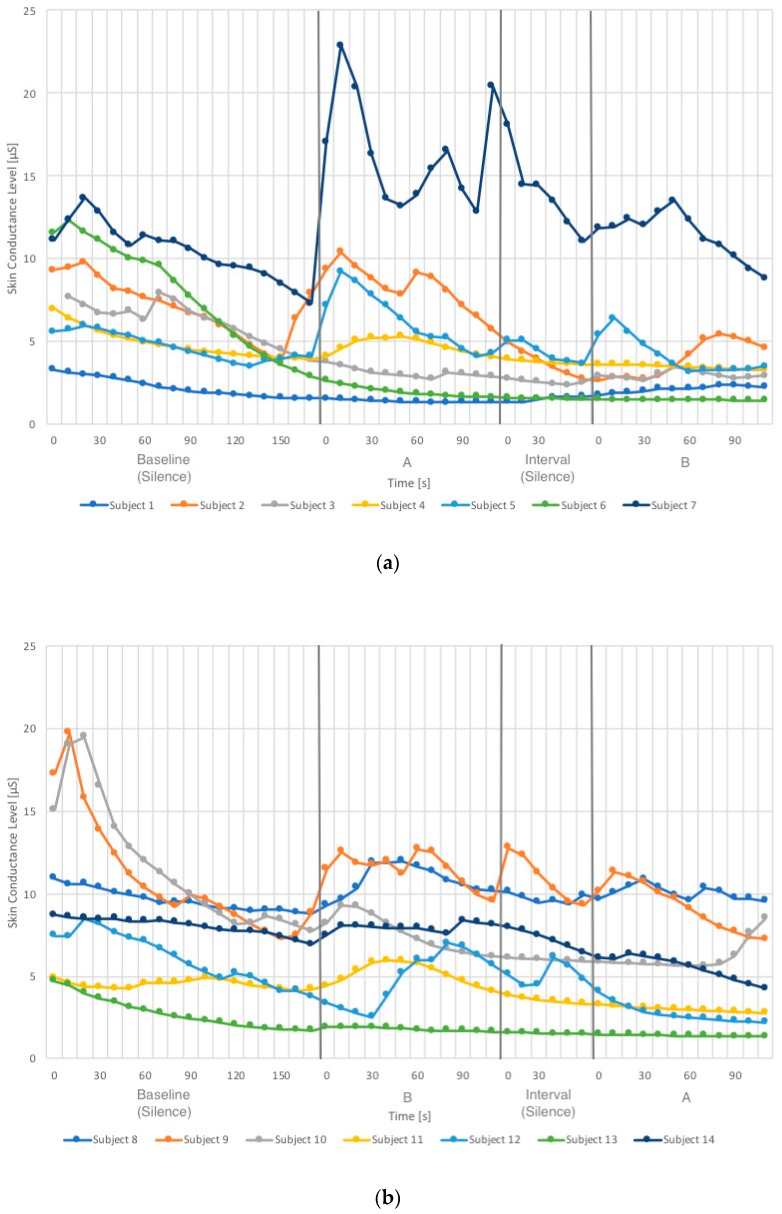
(**a**) The SCL (raw data) of Subjects 1 to 7 when “Birdsong without Car Noise” was presented first. (**b**) The SCL (raw data) of Subjects 8 to 14 when “Birdsong with Car Noise” was presented first. Higher rate of change of SCL corresponds to higher stress level. A: Birdsong without Car Noise; B: Birdsong with Car Noise. The vertical lines indicate the boundaries between the periods: the baseline, the first listening period (Sound A or B is played), the interval, and lastly the second listening period (Sound B or A is played).

**Figure 6 ijerph-16-04985-f006:**
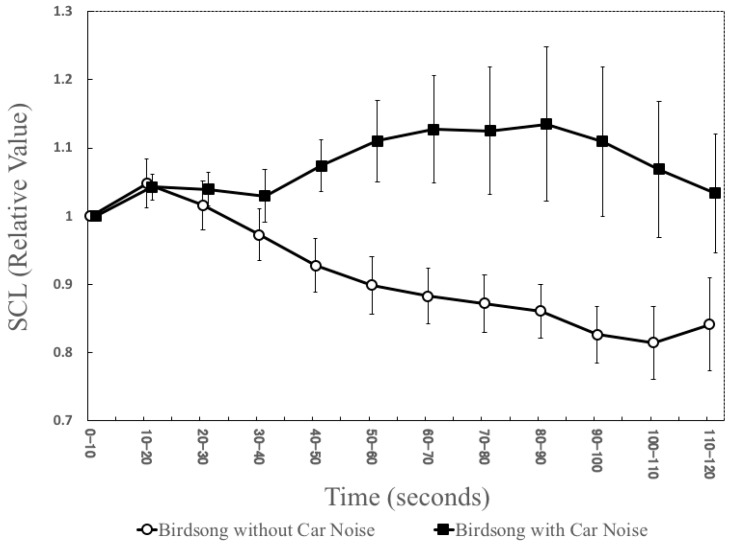
The relative values of the SCL for each sound. The graphs show the mean values and the standard errors of the preceding ten seconds across 14 participants. The SCL values were originally measured 500 times per second. A higher rate of change in SCL corresponds to a higher stress level.

**Table 1 ijerph-16-04985-t001:** The mean (± standard error (SE)) value of the Perceived Restorativeness Scale (PRS) score in each attribute under each condition.

PRS Attributes	CL	A	B	*p*-Value in Kruskal–Wallis Test
Being Away	6.40 ± 0.51	7.70 ± 0.54	5.93 ± 0.56	0.0588
Fascination	7.01 ± 0.48	8.00 ± 0.44	7.19 ± 0.43	0.2026
Coherence	4.77 ± 0.46	5.16 ± 0.41	4.75 ± 0.43	0.7844
Scope	3.43 ± 0.32b	**7.46 ± 0.42a**	**5.84 ± 0.59a**	0.0001 **
Compatibility	5.19 ± 0.44	6.44 ± 0.44	5.06 ± 0.39	0.0333 *
Familiarity	2.21 ± 0.92	3.50 ± 0.78	3.64 ± 0.72	0.1169
Preference	4.75 ± 0.55	6.50 ± 0.42	4.89 ± 0.56	0.0564
Total Score	5.29 ± 0.32b	**6.84 ± 0.34a**	5.64 ± 0.35ab	0.0105 *

CL; Control (Silence), A; Birdsong without Car Noise, B; Birdsong with Car Noise. Different letters after the SE in each column show a significant difference between the conditions (post-hoc Steel–Dwass Test) at *p* < 0.05 level. The significantly highest mean values in row are in bold. *n*: women = 6, men = 8. * *p* < 0.05, ** *p* < 0.01 between the conditions with Kruskal–Wallis test.

**Table 2 ijerph-16-04985-t002:** The relative mean value of the SCL under each condition and the *p*-value in Welch’s *t*-test during each 10 s.

Time (S)	Conditions (Mean ± *SE*)		
	A	B	*p*-Value in Welch’s T-Test
10–20	1.05 ± 0.0361	1.04 ± 0.0189	0.903
20–30	1.02 ± 0.0360	1.04 ± 0.0248	0.597
30–40	0.973 ± 0.0378	1.03 ± 0.0390	0.306
40–50	0.927 ± 0.0394	**1.07 ± 0.0380**	0.0129 *
50–60	0.898 ± 0.0418	**1.11 ± 0.0593**	0.0077 **
60–70	0.883 ± 0.0413	**1.13 ± 0.0793**	0.0128 *
70–80	0.872 ± 0.0417	**1.12 ± 0.0935**	0.0235 *
80–90	0.861 ± 0.0395	**1.14 ± 0.114**	0.0364 *
90–100	0.826 ± 0.0416	**1.11 ± 0.110**	0.0276 *
100–110	0.814 ± 0.0537	**1.07 ± 0.0997**	0.0360 *
110–120	0.841 ± 0.0679	1.03 ± 0.0870	0.0934

A: Birdsong without Car Noise; B: Birdsong with Car Noise. The significantly highest mean values in row are in bold. *n*: women = 6, men = 8. * *p* < 0.05, ** *p* < 0.01 between the conditions (Welch’s *t*-Test) at *p* < 0.05 level.

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
