# Peer review of "Effect of Faint Road Traffic Noise Mixed in Birdsong on the Perceived Restorativeness and Listeners’ Physiological Response: An Exploratory Study"

_ijerph, 2019, doi:10.3390/ijerph16244985_

Round 1
Reviewer 1 Report
The authors investigated whether the traffic noise affects the perceived restorativeness and the physiological restorativeness of birdsong in low-stress condition by measuring the SCL(skin conductance level) and the PRS(perceived restorativeness scale). The obtained conclusion is the SCL measurement is more important than the subject PRS assessing for describing the characteristics of restorative natural sound. However, prior to publication, a number of issues should be considered:
1. Compared with previous studies, a control condition of silence was added in this paper for the reference of natural sound with and without anthropogenic noise. But the innovation of this paper needs to be more clearly demonstrated, and a brief introduction about the potential practical application of this study also needs to be given.
2. The mean age of the participants is 25.5 years (S.D. = 3.11), consisting of 8 males and 6 females. For a more specific consideration, such as gender and age difference, there needs to be more participants and a wider age span.
3. Table 1 in row 235, n: women = 6, men = 8, while Table 2 in row 284, n: women = 10, men = 10?
And the meaning of p-value needs to be explained in details.
4. How to choose the interval time? Why it is one minute? What about the effect of interval on the result of the SCL and the PRS?
5. In experiment protocol, firstly, wearing a sleeping mask and listening. Then removing the sleeping mask and listening to fill in the PRS. Why do you choose this way?
6. In figure 6, the standard errors are relatively large. Can you explain it?
7. In discussion, this study indicated that the SCL would be useful to detect the negative effect of background noise on the restorativeness of birdsong when the noise cannot be perceived subjectively. However, the mechanism of the effect of background noise on the restorativeness of the birdsong in terms of the SCL has not been explained in details.
Author Response
Thank you very much for providing important insights. You will find our responses to each of your points and suggestions below. We are grateful for the time and energy you expended on our behalf.
(The contents of the attachment are the same as what is written below)
- Your questions are Q1-1, Q1-2, …, and our responses are A1-1, A1-2, …
- In the manuscript, we indicated revised parts with our comments such as “This is an answer to Q1-1.”
- In the manuscript, Q2-1, Q2-2, … are the questions asked by the other reviewer.
- In the manuscript, we replaced Figure 1 with a new one because we had found a mistake in the previous figure. We apologize for that.
- We have had the manuscript proofread.
Your questions and our responses are as follows:
Q1-1. Compared with previous studies,a control condition of silence was added in this paper for the reference of natural sound with and without anthropogenic noise. But the innovation of this paper needs to be more clearly demonstrated.
A1-1. Thank you for your suggestion and comment. Measuring the PRS under the silent condition as control was innovative because it enabled to assess the degree of restorativeness under the silent condition as the basis for comparing the effect of road traffic noise mixed with birdsong on the restorativeness with that of birdsong itself (We added this description in row 215 of the manuscript).
By contrast, many preceding studies did not examine whether natural sounds change the restorativeness compared to the silence condition, although they compared the effect of different types of acoustic stimuli (e.g., Medvedev et al., 2015 compared the effect of birdsong with that of traffic noise).
Q1-2. A brief introduction about the potential practical application of this study also needs to be given.
A1-2. Thank you for your suggestion. The potential practical application of this study is as follows: The outcome of this exploratory study can also be applied in medical settings, where staff members, especially surgeons and surgical nurses, are subjected to stress every day. Measuring surgeons' or nurses' SCL during surgery would be useful to detect their stress levels due to auditory disturbances as their negative effects are not perceived subjectively. This detection can help in creating a more efficient environment in medical settings.
Q1-3. The mean age of the participants is 25.5 years (S.D. = 3.11), consisting of 8 males and 6 females. For a more specific consideration, such as gender and age difference, there needs to be more participants and a wider age span.
A1-3. Thank you for your comment. We are sorry we did not explain it clearly enough.
Some of the preceding studies with physiological measurements were based on experiments with around fifteen participants (e.g., 15 participants (four women and 11 men) in Horiuchi et al. [22]; 12 participants (12 men) in Lee et al. [23]). This study was also exploratory; we intended to explore the possibility of combining psychological and physiological measurements from the viewpoint of an exploratory approach. Therefore, we considered that the sample size of this study (14 participants) could also be adequate for the experimental design (we added this description in row 509 of the manuscript).
Of course, further studies with more respondents are needed to scrutinize this subject (we added this description in row 905 of the manuscript).
Also, this study did not consider the gender gap because we intended to know the outcome regardless of the gender gap (we added this description in row 513 of the manuscript).
As for the participants’ age bracket, we limited the age span to around 25 years old because controlling the age span was considered to be useful to compare the results with those of previous studies or to develop the concept of this study in the future (we added this description in row 514 of the manuscript).
Q1-4. Table 1 in row 235, n: women = 6, men = 8, while Table 2 in row 284, n: women = 10, men = 10? And the meaning of p-value needs to be explained in details.
A1-4. Thank you for pointing them out. The difference between the number of participants in Table 1 and 2 was just a mistake. We apologize for that, and corrected them (in row 836 of the manuscript).
As for the p-value in table 2, it means the probability of obtaining the observed results of a test, assuming that there is no difference in the mean values of the relative values of the SCLs between the two conditions (we added this description in row 784 of the manuscript).
Q1-5.How to choose the interval time? Why it is one minute? What about the effect of interval on the result of the SCL and the PRS?
A1-5. Thank you for your question. We are sorry we did not explain it clearly enough. The length of the interval was determined based on the temporal difference between stimulus onset and recovery from a phasic increase in the SCL. According to the Handbook of Psychophysiology [13], the temporal difference between stimulus onset and the point of 50 % recovery of a phasic increase in the SCL is around 30 seconds. Therefore, the interval in this experiment should be longer than 30 seconds to differentiate effects of the first sound from those of the other sound, but the interval should not be too long so as not to make participants feel bored. Thus, a one-minute length of interval could be considered as appropriate in this study (we added this description in row 531 of the manuscript).
However, there is still a possibility that the interval affected the change in the SCL because the experiment did not contain a stressful task before each listening period (we added this description in row 899 of the manuscript).
Q1-6. In experiment protocol, firstly, wearing a sleeping mask and listening. Then removing the sleeping mask and listening to fill in the PRS. Why do you choose this way?
A1-6. Thank you very much for your question. We apologize for not explaining the reason why we chose this way clearly. We did this because he/she had to remove the sleeping mask to use his/her eyes and read the PRS questionnaire items and answer them. The participant listened to the natural sounds in the same order as he/she listened during the precedent SCL measurement period while filling in the PRS; "Birdsong without Car Noise" was being played while he/she was filling in the PRS to evaluate this sound, and "Birdsong with Car Noise" was being played while he/she was filling in the PRS to evaluate this sound. We intended to compare the effect of the acoustic stimuli on the perceived restorativeness of the experimental place relatively in this protocol.
(We added this description in row 587 of the manuscript).
Q1-7. In figure 6, the standard errors are relatively large. Can you explain it?
A1-7. Thank you for pointing it out. The standard errors of the relative values of the SCL with "Birdsong with Car Noise" increased 50 seconds after the acoustic stimulus onset, whereas "Birdsong without Car Noise" did not. This difference may imply that this effect of car noise would appear at least 50 seconds after birdsong begins, but the degree of this effect would differ from person to person. Mainly, the individual difference of the allowable range of artificial noise would be more extensive than that of natural components.
We added this description in row 876 of the manuscript.
Q1-8. In discussion, this study indicated that the SCL would be useful to detect the negative effect of background noise on the restorativeness of birdsong when the noise cannot be perceived subjectively. However, the mechanism of the effect of background noise on the restorativeness of the birdsong in terms of the SCL has not been explained in details.
A1-8. Thank you for your comments. As we explained in A1-7, the result of the SCL measurement can be interpreted that the effect of faint background noise on the restorativeness of birdsong in terms of the SCL can appear at least 50 seconds after a participant begins to listen to birdsong. It implies that the duration of acoustic stimuli should be more than 50 seconds to examine the effect of background noise.
Therefore, several preceding studies which assessed the subjective restorativeness of an environment through an experiment with a less than 50-second presentation time for stimuli (e.g., 30-second sound excerpts were evaluated subjectively in Axelsson et al., 2010 [24]; eight-second sound excerpts were assessed subjectively and physiologically in Hume and Ahtamad, 2013 [6]) might have overlooked the effect of artificial components which did not appear immediately.
In that sense, we consider that the SCL would be useful to detect the negative effect of background noise on the restorativeness of birdsong when the effect cannot be assessed subjectively.
We added this description in row 890 of the manuscript.
We look forward to hearing from you regarding our submission. We would be glad to respond to any further questions and comments that you may have.
Yours sincerely,
Yasushi SUKO

Reviewer 2 Report
This study is interesting and offers possibilities for therapeutic interventions. The results of the SCL show some promise. The PRS however did not show any significance. My main concerns are:
the low sample size - making this an exploratory study - and it should be renamed as such with greater critical reflection of the methods and ways of improving them. The use of the PRS - I think it may be questioned to what extent this scale is applicable to soundscapes only. In the description of the dimensions of the scale several would appear to be associated with landscapes or scenes that can be perceived to contain depth and other aspects which are possibly not apparent with sound only. The authors should include a critical discussion of this possibility as part of their explanation of the differences between the two methods If the PRS can be justified then more discussion about the experimental protocol is needed - should the participants have been subjected to a stress test as is usually done and is it possible to test for PRS twice in succession without repeating a stress test? What would be the ideal sample size and the possibility of dividing a larger sample of respondents into two groups, one of which gets the birdsong and the other the birdsong plus traffic noise? This approach is normal in other experiments such as looking out of windows at different views.Therefore the paper would be of greater value as a presentation of an exploratory methodology with such issues as I have raised being central to the discussion of an experimental design.
The paper has a briefly stated objective and no research questions, which should be included.
Author Response
Thank you very much for providing important insights. You will find our responses to each of your points and suggestions below. We are grateful for the time and energy you expended on our behalf.
- Your questions are Q2-1, Q2-2, …, and our responses are A2-1, A2-2, …
- In the manuscript, we indicated revised parts with our comments such as “This is an answer to Q2-1.”
- In the manuscript, Q1-1, Q1-2, … are the questions asked by the other reviewer.
- In the manuscript, we replaced Figure 1 with a new one because we had found a mistake in the previous figure.
- We have had the manuscript proofread.
Your questions and our responses are as follows:
Q2-1. The use of the PRS - I think it may be questioned to what extent this scale is applicable to soundscapes only. In the description of the dimensions of the scale several would appear to be associated with landscapes or scenes that can be perceived to contain depth and other aspects which are possibly not apparent with sound only. The authors should include a critical discussion of this possibility as part of their explanation of the differences between the two methods.
A2-1. Thank you for your comment. As you say, some of the descriptions are associated with landscapes rather than soundscapes. However, the PRS was developed to measure the perceived restorativeness which one particular environment has, and therefore it is not necessarily specific to assessing landscapes. Although many eyesight-oriented studies have used the PRS, there have also been on-site studies in which the PRS was used to evaluate the perceived restorativeness of the environment which participants felt with the five senses, and then it would be acceptable to use the PRS in a hearing-oriented study.
Using the PRS enabled this study to examine the validity of its outcome by comparing the result with those of previous studies which used the PRS. Besides, this study aimed to investigate whether the perceived restorativeness of the environment with different acoustic stimuli differ on the presupposition that the stimuli of the other four senses (including the eyesight) are the same. Therefore, the result of this study - the acoustic stimuli of birdsong changed the perceived restorativeness of the environment, but that of faint road traffic noise did not - was compatible with our intention to use the PRS.
We added this description in row 848 of the manuscript.
Q2-2. If the PRS can be justified then more discussion about the experimental protocol is needed - should the participants have been subjected to a stress test as is usually done and is it possible to test for PRS twice in succession without repeating a stress test?
A2-2. Thank you for your comment. Yes, this is one of the limitations of this study. There is some possibility that birdsong with or without car noise restore the participants in different ways if the participants had been under stress before each listening period (we added this description in row 901 of the manuscript).
The same is true of the two successive PRS.However, we counterbalanced the presenting order of the acoustic stimuli in both of the PRS and the SCL measuring periods. Besides, the one-minute interval, whose duration could be considered as adequate to differentiate the effect of the first sound from that of the second sound, was provided between the two listening periods for the SCL measurements (we added this description in row 534 of the manuscript).
Therefore, we consider that the absence of a stress test would be acceptable in the framework of this exploratory study.
Q2-3. What would be the ideal sample size and the possibility of dividing a larger sample of respondents into two groups, one of which gets the birdsong and the other the birdsong plus traffic noise? This approach is normal in other experiments such as looking out of windows at different views.
A2-3. Thank you for your question. Yes, as you say, dividing a larger sample of respondents into two groups is a widely used approach (e.g., Ulrich, 1984. View through a window may influence recovery from surgery. Science, 224(4647), 420–421), but in the study by Ulrich (1984), the data from the two groups did not correspond to each other.
This study intended to compare paired data (birdsong with/without car noise), which was obtained from the same person. We counterbalanced the order effect of the acoustic stimuli with changing the presentation order. Many preceding studies whose experimental protocols were similar to that of this study counterbalanced the order effect in such a way. Regarding the sample size (14 respondents), we followed Horiuchi et al. [22] and Lee et al. [23], and we considered it as adequate.
We added this description in row 510 of the manuscript.
Q2-4. The paper has a briefly stated objective and no research questions, which should be included.
A2-4. Thank you very much for pointing it out. We apologize for that, and we added the following research question at the end of the introduction part (Row 223, page 2):
“Does faint road traffic noise mixed in birdsong affect the restorativeness of the birdsong in terms of physiological evaluation even if the noise does not influence the perceived restorativeness of the place of experiment where people are exposed to the birdsong?”
We look forward to hearing from you regarding our submission. We would be glad to respond to any further questions and comments that you may have.
Round 2
Reviewer 2 Report
The authors have taken all my review points into account and also renamed the study as an exploratory one, which I believe is very appropriate. I am now happy to endorse it for publication